# Exploring the Influence of Growth-Associated Host Genetics on the Initial Gut Microbiota in Horses

**DOI:** 10.3390/genes14071354

**Published:** 2023-06-27

**Authors:** Jongan Lee, Yong-Jun Kang, Yoo-Kyung Kim, Jae-Young Choi, Sang-Min Shin, Moon-Cheol Shin

**Affiliations:** 1Subtropical Livestock Research Institute, National Institute of Animal Science, RDA, Jeju 63242, Republic of Korea; yjkang1201@korea.kr (Y.-J.K.); moolmaru@korea.kr (Y.-K.K.); jaechoi@korea.kr (J.-Y.C.); adamrib@korea.kr (S.-M.S.); 2Planning and Coordination Division, National Institute of Animal Science, RDA, Wanju 55365, Republic of Korea; shinemoon@korea.kr

**Keywords:** horse, gut microbiota, SNP, genotype, growth

## Abstract

The influences of diet and environmental factors on gut microbial profiles have been widely acknowledged; however, the specific roles of host genetics remain uncertain. To unravel host genetic effects, we raised 47 Jeju crossbred (Jeju × Thoroughbred) foals that exhibited higher genetic diversity. Foals were raised under identical environmental conditions and diets. Microbial composition revealed that *Firmicutes*, *Bacteroidetes*, and *Spirochaetes* were the predominant phyla. We identified 31 host–microbiome associations by utilizing 47,668 single nucleotide polymorphisms (SNPs) and 734 taxa with quantitative trait locus (QTL) information related to horse growth. The taxa involved in 31 host–microbiome associations were functionally linked to carbohydrate metabolism, energy metabolic processes, short-chain fatty acid (SCFA) production, and lactic acid production. Abundances of these taxa were affected by specific SNP genotypes. Most growth-associated SNPs are found between genes. The rs69057439 and rs69127732 SNPs are located within the introns of the *VWA8* and *MFSD6* genes, respectively. These genes are known to affect energy balance and metabolism. These discoveries emphasize the significant effect of host SNPs on the development of the intestinal microbiome during the initial phases of life and provide insights into the influence of gut microbial composition on horse growth.

## 1. Introduction

The gut microbiota in the gastrointestinal tract (GIT) maintains a symbiotic linkage to the host and positively contributes to nutrient absorption and immune system regulation, but it also contributes adversely to the onset of metabolic diseases [1,2,3]. The horse’s GIT can be divided into two main sections: the foregut and the hindgut. Microbial fermentation of dietary fiber occurs in the hindgut, generating volatile fatty acids (VFAs), including acetate, propionate, and butyrate, which contribute to the daily energy requirements for growth and maintenance [4]. Diverse commensal microbial communities, including bacteria, archaea, fungi, parasites, and protozoa, are present in the GIT of horses [5]. Foals possess an abundant and assorted microbial community, with *Firmicutes* being the predominant phylum [6]. However, the microbiome transitioned after the first 60 days and attained relatively stable microbial communities.

While it is generally recognized that diet affects the makeup of the gut microbial communities [7,8], emerging evidence suggests that host genetics also serve as crucial determinants [9,10]. Quantitative trait locus (QTL) mapping in a mouse model discovered segments of the mouse genome that are linked to *Akkermansia muciniphila* abundance [11]. Subsequent experiments have shown that *A. muciniphila*, a main contributor of ornithine lipids within the gastrointestinal system, supports its immunomodulatory effects. In another study, genome-wide significant variants and heritability estimates of bacteria yielded initial evidence of the interplay between cattle genotypes and skin bacteria of the foot [12]. Nevertheless, a few studies have found that diet and environmental conditions influence gut microbiota structure more significantly than host genetics [13,14]. The discordance in understanding the impact of host genetics and other components may be due to variations in environmental conditions, growth phases, and populations.

The commensal gut microbiome of horses is crucial for optimal feed efficiency, including nutrient use, inflammation regulation, immune homeostasis, and energy metabolism, to support growth [15]. In horses, the rapid growth period, similar to the infantile growth phase in humans, lasts from birth to weaning and typically occurs at 4–6 months of age [16]. Hence, properly establishing and maintaining nutrition-related microbiota in the GIT is crucial for horse growth during the developmental phase. Interestingly, early colonization of the swine gut microbiome by *Prevotella* was positively related to body weight [17]. Additional research revealed that the expression of rumen bacteria in the *Firmicutes* phylum affects the growth performance of calves [18]. Significantly, variations in the host’s genes that control the immune system and growth affect the abundance of the gut microbiota [19]. A study on a chicken F2 resource population found that a single nucleotide polymorphism (SNP) rs15142674 located in the *PLAG1* and *LYN* genes associated with cell specialization and development regulates specific methanogens [20].

Based on these findings, we hypothesized that horse populations with distinct genetic variations in growth traits would show visual impacts on the gut microbiota composition, influencing nutrient metabolism. We selected Jeju crossbred (Jeju × Thoroughbred) horses with high genetic diversity to identify the contribution of host genetics to the early development of the gut microbiome. To minimize external factors, excluding host genetic effects, foals were raised under the same circumstances and dietary conditions. We subsequently analyzed the gut microbial composition of foals aged 4–7 months before weaning. We focused on horse growth-associated QTL and investigated growth-associated variants that influence the shaping of the early gut microbiome. This research is the first investigation to uncover the influence of host SNPs on the gut microbial community in horses and provides valuable insights into this previously unexplored relationship.

## 2. Materials and Methods

### 2.1. Animals and Sample Collection

All experimental animal procedures were approved through the Institutional Animal Care and Use Committee (IACUC) at the National Institute of Animal Science (NIAS) (approval number: NIAS20212145). A total of 47 Jeju crossbred (Jeju × Thoroughbred) foals, comprising 19 males and 28 females, were used in this study. The foals were 4–7 months old and in the preweaning phase. After birth, the foals were raised with their dams in individual horse stables for one month at the Subtropical Livestock Research Institute in Korea. The foals were reared in groups with their dams in a larger horse stable from two to seven months of age. The foals were provided with clean water, orchard grass hay, and foal starter (pellets) to supplement their diets. At three months of age, the strangles vaccine was administered to prevent infectious diseases that could be transmitted through group rearing. At six months of age, foals were vaccinated against equine influenza and herpes viruses to prevent respiratory diseases. A veterinarian periodically assesses the health of foals. Complete blood count (CBC) tests were performed every two months, and foals with an increased white blood cell count significantly above the standard were administered antibiotics and anti-inflammatory drugs to prevent infectious diseases. The foals were generally healthy from birth to seven months of age. Fecal and blood samples were collected from the 47 foals. Fresh specimens of feces were collected from the rectum using disposable gloves and sampling tubes (SPL, Seoul, Korea) to minimize environmental contamination. The collected fecal samples were promptly frozen in a deep freezer at −70 °C. Jugular blood specimens (5 mL per foal) were collected from the foals and stored at 4 °C.

### 2.2. 16S rRNA Amplicon Sequencing

Genomic DNA (gDNA) was isolated from a frozen stool specimen (250 mg) using the QIAamp PowerFecal Pro DNA kit (Qiagen, Hilden, Germany). The concentration and purity of the extracted gDNA (A260/A280 and A260/A230) were evaluated using a NanoDrop ND-2000 spectrophotometer (Thermo Scientific, Waltham, MA, USA), and the gDNA were kept at −20 °C. Libraries for DNA sequencing were set up following Illumina 16S Metagenomic Sequencing Library protocols, especially targeting the magnification of the V3–V4 hypervariable segments of the 16S ribosomal RNA gene. The quantification of the final purified products was conducted using qPCR, following the qPCR Quantification Protocol Guide provided by KAPA Library Quantification kits designed for the Illumina Sequencing System. Paired-end (2 × 300 bp) reads were obtained using the Illumina MiSeq System (Illumina, San Diego, CA, USA).

### 2.3. Animal Genotyping

DNA was extracted from blood specimens using a G-DEX IIb Genomic DNA Extraction kit (iNtRon Biotechnology, Seongnam-si, Korea), following the manufacturer’s guidelines. The extracted DNA samples were genotyped using the Illumina GGP Equine SNP70 BeadChip, which contains 65,157 SNPs. GenomeStudio 2.0 software (Illumina, San Diego, CA, USA) was utilized to identify the genotypes of 65,157 SNPs. The PLINK input report plug-in (v2.14) of GenomeStudio was used to generate PED and MAP files. The quality control (QC) procedure was executed using PLINK [21], which included filtering criteria such as minor allele frequency (<5%), percentage of genotype call (<95%), and Hardy–Weinberg equilibrium (HWE) deviations (chi-square test, *p* < 10^−6^). Following the QC procedure, 17,489 SNPs were removed from the dataset because they did not meet the filtering criteria. The remaining 47,668 SNPs were used for downstream analyses.

### 2.4. Taxonomic Assignment and Diversity

QIIME2 (release 2020.11) was employed for the taxonomic assignment of 16S amplicon sequences [22]. The Cutadapt tool in QIIME2 removed forward and reverse primer sequences. The DADA2 tool was used to truncate the sequencing reads (parameters: p-trunc-len-f 283 and p-trunc-len-r 206) to ensure data quality and mitigate the impact of sequencing errors. Amplicon sequencing variants (ASVs) for feature identification were obtained through the DADA2 denoising algorithm. The SILVA (release 132) reference database and the q2-feature-classifier module were used to designate taxonomic classifications for the ASVs. The α diversity between samples was assessed using the q2-diversity plug-in. The gut microbiota’s richness (Chao1), evenness (Simpson and Shannon), and completeness (Good’s coverage) were evaluated through the α diversity analysis.

### 2.5. Growth-Associated Host Genetic Effects on Gut Microbiota

The correlation between SNP genotypes and gut microbial abundance was assessed using the Matrix eQTL R package (v2.1) [23]. The relative abundances of taxa were transformed using an arcsine function to normalize their distribution. Linear regression analysis was conducted using the modelLINEAR function, with the SNP genotype matrix (0: two copies of the reference allele, 1: one copy of the alternative allele, 2: two copies of the alternative allele, NA: not applicable) as independent variables and the relative expressions of each taxon as the dependent variable. These parameters were incorporated as covariates to address the potential confounding effects of age (days) and sex (0, male; 1, female). The false discovery rate (FDR) value was obtained using the Benjamini–Hochberg method during Matrix eQTL analysis. Associations with FDR values below 0.1 were deemed significant, while associations falling within the range of 0.1–0.25 were considered to be suggestively significant. To select growth-associated SNPs from the identified associations in the Matrix eQTL analysis, the growth-related 616 QTL information in the Horse QTLdb (Equcab2.0) was used [24]. Following this procedure, we narrowed our focus to genetic variants previously associated with growth traits in horses. In the outcomes of the host–microbiome association, SNPs were annotated using Equcab2.0 (Ensembl release 94). Concurrence with horse growth-associated taxa was determined using the ADDAGMA database [25]. A plot of the genomic arrangements was created using a phenogram [26].

## 3. Results

### 3.1. Gut Microbial Composition and α Diversity

The taxonomic composition was categorized into 23 phyla, 42 classes, 84 orders, 179 families, 441 genera, and 707 species using SILVA (release 132). Among the taxa, one class, five orders, 44 families, 126 genera, and 566 species are classified as uncultured or ambiguous taxa. The relative abundance, sample prevalence, and standard deviation of the taxa are presented in Appendix A. Taxa with a sample prevalence exceeding 20% included 21 phyla, 36 classes, 57 orders, 110 families, 235 genera, and 265 species. The dominant phyla were *Firmicutes* (45.4%), *Bacteroidetes* (28.5%), and *Spirochaetes* (9.8%) (Figure 1A). The genera *Treponema 2* (9.1%), *Streptococcus* (8.6%), and *Rikenellaceae RC9 gut group* (8.0%) showed higher expression (Figure 1B). *Firmicutes* exhibited the most significant variation in abundance among phyla, whereas *Streptococcus* displayed the highest standard deviation among genera (Appendix A).

We performed α diversity analysis to identify the distribution of microbes within a sample by estimating richness and evenness (Figure 1C). In all samples, the Good’s coverage index was 99% or higher, indicating that the depth of sequencing adequately encompassed the full spectrum of the gut microbiota. Chao1, an indicator of species richness, was used to measure the total species count in the sample. The Chao1 values across the samples ranged between 756 and 1824, indicating significant variability in the number of species. Species richness and evenness were assessed using the Shannon and Simson indices, with 7.01 to 9.67 and 0.92 to 0.99, respectively. The median values of the Chao1, Shannon, and Simpson indices were 1150, 8.74, and 0.9922, respectively. Diversity analysis results indicated notable differences in the distribution of microbes among the samples.

### 3.2. Interplay between Growth-Associated Variants and Gut Microbiota

For Matrix eQTL analysis, uncultured and ambiguous taxa were excluded. We used a total of 734 taxa, comprising 23 phyla, 41 classes, 79 orders, 135 families, 315 genera, and 141 species. A total of 47,668 SNPs (Appendix A) and covariates (Appendix A), including age and sex, were used in the analysis. Matrix eQTL analysis yielded 404,196 associations (*p* < 0.01). After applying additional filtering criteria, a subset of 4227 associations was identified. The criteria were that the microbiota sample prevalence was greater than 20% and the FDR was less than 0.25. Upon mapping the 4227 associations to growth-associated QTL in horse QTLdb, 31 host–microbiome interactions related to horse growth traits were identified (Figure 2A). Eighteen taxa identified within 31 associations corresponded to horse growth-related taxa in the ADDAGMA database (Appendix A). Among the 31 identified associations, two were found to be statistically significant (FDR < 0.1), while the remaining 29 associations were considered to be suggestively significant (0.1 < FDR < 0.25). Appendix A presents details of the 31 associations. Among the 31 associations, 6 SNPs (rs68704438, rs68550859, rs68752065, rs68952503, rs69057439, and rs69127732) were correlated with multiple taxa.

In our study, we observed four associations with metabolic processes. The A allele of the rs68704438 SNP has been linked with a higher expression of *Coriobacteriales* involved in metabolizing diverse carbohydrates [27] (Figure 2B). The expression levels of *Methanobrevibacter* positively correlated with the T allele of the rs68550859 SNP (Figure 2C). *Methanobrevibacter* affects animals’ energy metabolism and growth [28,29]. The rs68952503 SNP was associated with decreased *Desulfovibrio* abundance (Appendix A). *Desulfovibrio* metabolizes both energy and carbon [30]. In addition, *Succinivibrionaceae*, which is associated with feed efficiency and animal productivity [31], exhibited distinct expression patterns depending on the rs69057439 SNP (Appendix A). rs69057439 is located within the intron of the *VWA8* gene and is involved in energy metabolism [32].

Among the 31 associations, we identified two taxa engaged in short-chain fatty acid (SCFA) production and four related to lactic acid production and utilization. As shown in Figure 2D, the G allele of the rs68837751 SNP is linked to reduced expression levels of *Schwartzia*, contributing to propionate production [33]. Furthermore, the expression levels of *Mollicutes RF39*, recognized for its role in enhancing SCFA production [34], were influenced by the rs69127732 SNP (Appendix A). rs69127732 is located within the intron of the *MFSD6* gene and is implicated in regulating energy balance [35]. Notably, the relative abundances of *Streptococcus*, known as lactic acid bacteria (LAB) [36], the lactate-producing taxa *Atopobiaceae* and *Catenisphaera* [37,38], and the lactate-utilizing bacterium *Tepidimicrobium* [39], varied depending on the effective alleles of the SNPs.

## 4. Discussion

The gut microbiota of horses is widely recognized to be predominantly affected by environmental components such as dietary patterns and rearing conditions, leading to variations in microbial composition and expression levels [40,41]. The foals were raised under identical diets and environmental conditions to minimize environmental factors. We expanded the host genetic spectrum to assess alterations in gut microbiota by utilizing Jeju crossbred horses (Jeju × Thoroughbreds) with higher SNP heterozygosity than Jeju horses and Thoroughbreds [42]. Furthermore, we specifically targeted preweaning foals aged 4–7 months because this period represents horses’ most critical growth phase. This study examined how genetic variants associated with growth affect specific microbial populations.

The fecal microbiota of Jeju crossbred foals exhibited a similar microbial composition, with *Firmicutes*, *Spirochaetes*, and *Fibrobacteres* representing the majority of the phyla, to that of Quarter Horse foals aged 4–6 months [43]. During the weaning period of thoroughbred horses, the Shannon index ranged from 7.3 to 7.8, indicating relatively minimal variation in diversity compared to our findings [44]. The evidence from these results suggests that the genetic influence on the intestinal microbiome community is not restricted to specific horse breeds but has broader applicability across diverse breeds. A comprehensive database, ADDAGMA, was recently developed to explore the gut microbiomes of domestic animals with various traits. Among the microbiota associated with horse growth traits in ADDAGMA, we discovered 18 taxa in 31 host–microbiome associations. The 18 identified taxa were distributed as follows: four in *Firmicutes*, six in *Proteobacteria*, two in *Bacilli*, one in *Coriobacteriia*, three in *Coriobacteriales*, and two in *Veillonellaceae*. These findings suggest that host genetics and the microbiome interact within the growth phenotype.

Supplying adequate nutrients during growth is crucial to horse growth [45]. Carbohydrates have been suggested to influence bone development potential and are considered vital nutrients required for foal development [46]. *Coriobacteriales* have also been reported to participate in glucose fermentation and diverse carbohydrates [47]. The expression levels of *Coriobacteriales* changed depending on the rs68704438 genotype. Energy metabolism produces energy from nutrients and is closely associated with animal health and growth [48]. *Methanobrevibacter* and *Desulfovibrio*, which participate in energy metabolism, exhibited alterations in their abundance according to the genotypes of SNPs rs68550859 and rs68952503, respectively. The continuous supply of SCFA from dietary fiber can meet the daily energy requirements, ensuring consistent metabolite availability and stable hormone levels in horses [49]. The expression levels of *Schwartzia* and *Mollicutes RF39*, known to increase SCFA production, were influenced by the SNPs rs68837751 and rs69127732, respectively. rs68704438, rs68550859, rs68952503, and rs68837751 were associated with withers height in the Horse QTLdb, while rs69127732 was related to body weight. Lactate has traditionally been misunderstood as a metabolic waste product primarily associated with the onset of fatigue during exercise. Recent studies have revealed its involvement in regulating various aspects of energy metabolism and signal transduction [50]. Furthermore, lactate plays a dual role in brain health, functioning as both an energy provider and a signaling agent [51]. Our results revealed that SNP genotypes influenced four taxa associated with lactate production: *Streptococcus*, *Atopobiaceae*, *Catenisphaera*, and *Tepidimicrobium*. Nevertheless, owing to the current limited understanding of the specific functions of lactate as an energy source in horses, further research is necessary to fully understand its role.

In summary, this study unveiled the connections between host SNPs and gut microbial composition in horses. Using horse growth-related QTL information, we discovered 31 host–microbiome associations related to horse growth. The taxa identified in these associations play a role in carbohydrate and energy metabolism, SCFA production, and lactate production. Additional research is required to uncover the effect of other components, including breed, sex, and growth phase, on microbial communities beyond host genetics. Understanding the mechanisms governing the stability and dynamics of the gut microbiome environment is vital to improving animal health and productivity.

## Figures and Tables

**Figure 1 genes-14-01354-f001:**
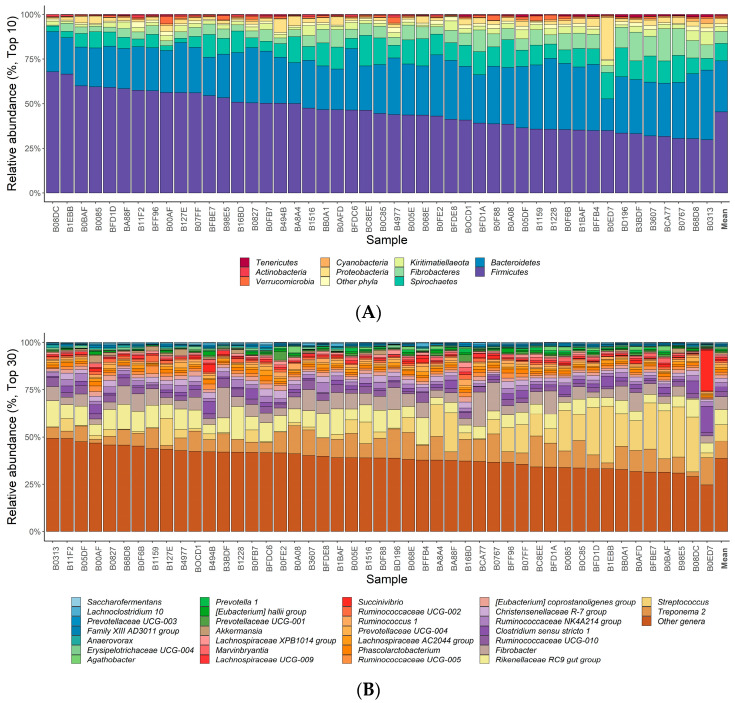
Relative abundances (%) of microbiota and α diversity indices were obtained from 47 Jeju crossbred foals’ fecal samples. (**A**) Bar chart of the ten most abundant phyla. (**B**) Bar chart of the 30 most abundant genera. (**C**) Boxplot of α diversity indicators. The first quartile (Q1) is at the bottom of boxes, and the third quartile (Q3) is at the top of boxes. The second quartile (Q2), which represents the median value, is represented by a thick line within the box. The red, blue, green, and yellow dots in the boxplots represent observed values of α diversity indicators for each sample.

**Figure 2 genes-14-01354-f002:**
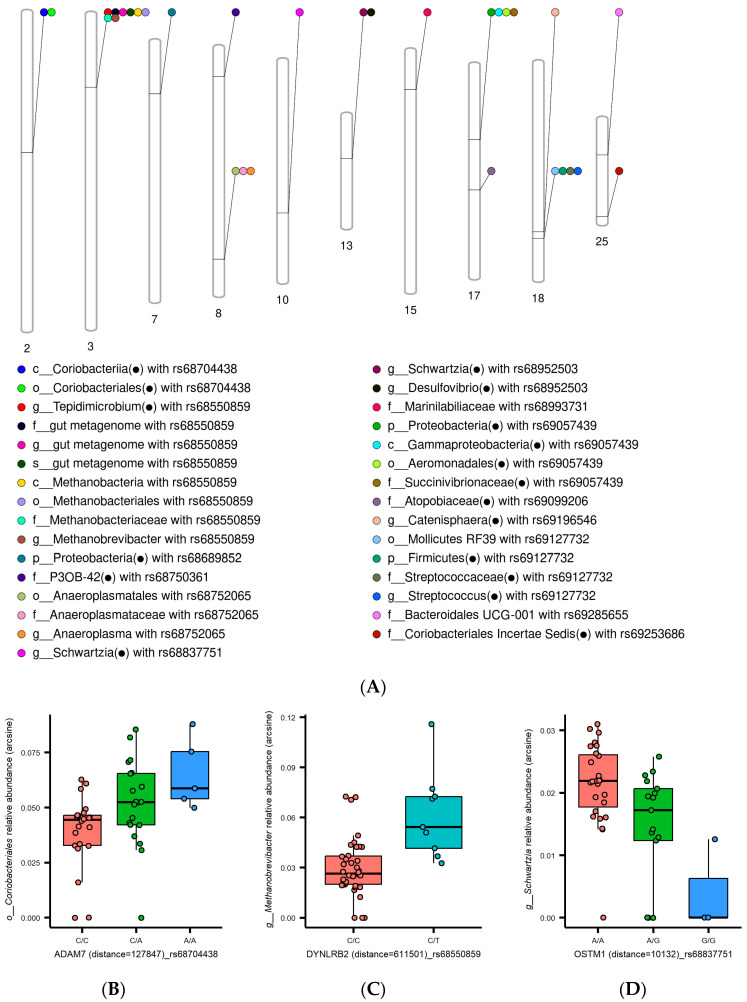
Interactions between growth-related variants and gut microbiota. (**A**) The plot of genomic arrangement shows the 31 host–microbiome associations on horse chromosomes 2, 3, 7, 8, 10, 13, 15, 17, 18, and 25. The dark circle (●) denotes horse growth-related taxa in the ADDAGMA database. (**B**) *Coriobacteriales* relative abundance of the rs68704438 genotype (C: reference allele, A: alternative allele). (**C**) *Methanobrevibacter* relative abundance of the rs68550859 genotype. (C: reference allele, T: alternative allele) (**D**) *Schwartzia* relative abundance of the rs68837751 genotype (A: reference allele, G: alternative allele). The boxplots illustrate the differences in relative abundances of taxa according to the SNP genotype.

## Data Availability

The corresponding author will provide the experiment data upon reasonable request.

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
