# Peer review of "Exploring the Influence of Growth-Associated Host Genetics on the Initial Gut Microbiota in Horses"

_genes, 2023, doi:10.3390/genes14071354_

Round 1
Reviewer 1 Report
The authors touched on an important and interesting topic which is the relationship between the gastrointestinal microbiome and host genetics.
I find the manuscript interesting . However, I have a minor comments to the authors:
- it would be useful to briefly describe the conditions for maintaining the animals
- was the health status of animals monitored, if so what parameters were analyzed in this scope?
Reviewer 2 Report
The authors studied the influence of growth-associated host genetics on the initial gut microbiota in horses. They found that the host genetics played a significant role in shaping the gut microbiota during the initial phases of life.
I have following questions or suggestions to the authors.
1. Why did the authors only select the limited number of SNPs related with growth traits to analyze the host genetics’ impact on gut microbiota composition? Did the authors conduct GWAS using the genome-wide SNPs data and the gut microbiota composition? If yes, what did the authors find? Did the results of the GWAS verify the findings in the manuscript?
2. Gut microbiota composition is affected by both host genetics and the environmental factors just as the authors mentioned. According to the study, how much variation of the gut microbiota composition could be explained by the host genetics?
The authors should have a careful check to their writing. For example, the number “1842” and “1150” in Line 171 and 174, respectively , should be rewritten with as “1,842” and “1,150”.
The English writing is good. But additional check is recommended.
